# Robust Inter-Vehicle Distance Measurement Using Cooperative Vehicle Localization

**DOI:** 10.3390/s21062048

**Published:** 2021-03-14

**Authors:** Faan Wang, Weichao Zhuang, Guodong Yin, Shuaipeng Liu, Ying Liu, Haoxuan Dong

**Affiliations:** School of Mechanical Engineering, Southeast University, 2 Southeast University Road, Jiangning District, Nanjing 211189, China; faanwang@seu.edu.cn (F.W.); wezhuang@seu.edu.cn (W.Z.); spl@seu.edu.cn (S.L.); ying940303@seu.edu.cn (Y.L.); donghx@seu.edu.cn (H.D.)

**Keywords:** connected and automated vehicle, GNSS, V2X, inter-vehicle distance, pseudorange, cooperative localization

## Abstract

Precise localization is critical to safety for connected and automated vehicles (CAV). The global navigation satellite system is the most common vehicle positioning method and has been widely studied to improve localization accuracy. In addition to single-vehicle localization, some recently developed CAV applications require accurate measurement of the inter-vehicle distance (IVD). Thus, this paper proposes a cooperative localization framework that shares the absolute position or pseudorange by using V2X communication devices to estimate the IVD. Four IVD estimation methods are presented: Absolute Position Differencing (APD), Pseudorange Differencing (PD), Single Differencing (SD) and Double Differencing (DD). Several static and dynamic experiments are conducted to evaluate and compare their measurement accuracy. The results show that the proposed methods may have different performances under different conditions. The DD shows the superior performance among the four methods if the uncorrelated errors are small or negligible (static experiment or dynamic experiment with open-sky conditions). When multi-path errors emerge due to the blocked GPS signal, the PD method using the original pseudorange is more effective because the uncorrelated errors cannot be eliminated by the differential technique.

## 1. Introduction

Connected and automated vehicles (CAVs) promise many benefits for future mobility, including reducing traffic congestion, enhancing vehicle safety and improving energy efficiency of transportation system [1,2,3]. As an essential function of CAVs, robust and accurate localization has been widely studied in recent years.

In general, the accuracy requirements for CAV localization can be divided into four levels, i.e., road-level, lane-level, decimeter-level and centimeter-level. Road-level localization provides basic applications or services due to its rough positioning, for example, store or hospital navigation [4]. Lane-level localization can benefit more driving situations, further improving driving safety by lane-selection, and reduce energy consumption by eco-driving [5]. Decimeter-level localization is required for L2-L3 advanced driving assistance system (ADAS) applications [6], which are related to vehicle driving safety, e.g., active obstacle avoidance. A fully autonomous vehicle may require higher position estimation accuracy (centimeter-level) because of the complex driving conditions to ensure it stays in its lane or a safe distance from other vehicles.

The most common vehicle localization method is the global navigation satellite system (GNSS), which estimates the vehicle location from pseudorange measurements of multiple satellites. The existing errors of the pseudorange, for example, satellite clock error, ionospheric delay and multipath error, make GNSS fulfill only the road-level localization applications [7]. To achieve higher positioning accuracy for CAVs, a variety of techniques have been proposed, including combining measurements of additional sensors (Inertial Measurement Unit) [8], differential GNSS techniques (Real Time Kinematic) [9], Cooperative Map Matching (CMM) [10], Simultaneous Localization and Mapping (SLAM) [11] and so on.

In addition to single-vehicle localization, some safety-critical CAV applications, e.g., vehicle platoon, require accurate measurement of the inter-vehicle distance (IVD) [12,13,14]. The simplest IVD estimation method is differencing the vehicles’ position directly. However, its accuracy is highly dependent on the localization accuracy of a single vehicle. Using onboard millimeter-wave radar and lidar is another approach, but its perception range is limited [15].

Recently, cooperative localization methods have been proposed as alternative solutions to improve the positioning accuracy by sharing localization information between two or more sources (vehicles and infrastructures) through emerging vehicular communication technologies [16,17,18,19]. There are two typical cooperative localization methods, i.e., ranging-based and non-ranging based. The ranging-based methods use the signal strength variation or the transmission time to estimate the IVD, including Radio Signal Strength (RSS) [20], Time of Arrival (TOA) [21], Round Time Trip (RTT) [22] and Time Difference of Arrival (TDOA) [23,24,25]. However, these techniques usually require additional hardware or pre-deployed infrastructures, which may incur more cost. In addition, the high vehicle speed may also introduce noise or errors to the estimated distances.

The non-ranging cooperative localization method is a cost-effective solution that directly uses the pseudorange information of each vehicle to estimate the IVD. However, as the vehicles are moving relative to each other, the low estimation accuracy of the cooperative localization restrains its application in reality. Some studies have used the developed wireless communication techniques, e.g., Dedicated Short Range Communications (DSRC), for pseudorange exchange between vehicles [26]. The exchanged pseudorange information is used for IVD estimation by applying multi-source fusion [27,28,29]. Richter and Liu both proposed the double-differencing framework to measure the vehicle relative distance [27,28]. Richter used the particle filter to remove common GNSS errors from the pseudoranges of both vehicles [27], while Liu used the weighted least squares method [28]. As a result, the estimation accuracy of the IVD is improved. Another study presented by Golestan et al. proposed a multi-source fusion method to improve the accuracy of IVD measurement by combining different positioning technologies [29]. Tomic employed the maximum likelihood convex optimization method [30], and Naseri used the Bayesian estimation method to improve the accuracy of the distance between two points [31]. In addition, Guo achieved an infrastructure-free cooperative relative localization by using an onboard ultra-wideband ranging and communication network [32]. However, the literature mentioned above all assumed that GNSS errors were small and that no multipath error existed. Tahir et al. proposed several range measurement methods, including single and double differencing. The accuracy of the proposed methods is compared by actual field trails in different mobile environments [33]. In addition, Ansari also proposed a DSRC-based Vehicle-to-vehicle (V2V) real-time relative localization method and investigated the benefits of the proposed method [19]. However, no experiments were carried out to verify the effectiveness of the proposed methods.

Therefore, in this paper, we explore several non-ranging cooperative localization methods to estimate the IVD for a group of connected vehicles, including Absolute Position Differencing (APD), Pseudorangs Differencing (PD), Single Differencing (SD) and Double Differencing (DD). The main contributions of this paper are twofold. First, four different IVD estimation frameworks are formulated and compared. The weighted least squares method is employed to reduce the pseudorange errors and noises of each vehicle. The correlation errors of pseudoranges (i.e., satellite clock error, satellite ephemeris error, ionospheric error and tropospheric error) are greatly reduced. Second, field experiments, including static and dynamic, open-sky and GNSS-blocked driving scenarios, were conducted to verify their effectiveness. Among these methods, DD indicated the highest IVD measurement accuracy in open sky conditions, while PD showed the best accuracy in urban driving conditions with shelter.

This paper is organized as follows. Section 2 introduces the main errors of pseudorange and formulates the problem. In Section 3, four IVD estimation methods are presented by using the pseudorange of each vehicle. Experimental Results and Discussion are given in Section 4. Section 5 concludes this paper.

## 2. Problem Formulation of Inter-Vehicle Distance Measurement

This paper focuses on inter-vehicle distance measurement. The non-ranging cooperative localization method is a promising approach to achieve accurate IVD estimation. Figure 1 shows the concept of cooperative positioning by using multi-source information fusion. The vehicles and infrastructures can communicate with each other and share their positions through V2X techniques. Note that since the signal of GNSS may be lost in the environments of building blockings and tunnels, the critical situations that can severely reduce the complement of GNSS signal are not considered in this paper.

This paper mainly uses GPS observations to estimate the IVD between vehicles. At any time t, the pseudorange $\rho_{V}^{S}\left(t\right)$ from vehicle $V\in \left\{v_{1},v_{2},\cdots ,v_{n}\right\}$ to the receiving satellite S can be modeled as [26]
(1)$$\rho_{V}^{S}\left(t\right)=R_{V}^{S}\left(t\right)+t_{V}^{S}\left(t\right)+\varepsilon_{c}\left(t\right)+\varepsilon_{u}\left(t\right)$$
where $R_{V}^{S}\left(t\right)=∥L_{S}\left(t\right)-L_{V}\left(t\right)∥$ is the true distance between vehicle V and satellite S, $L_{S}\left(t\right)={\left[\begin{array}{ccc} x_{S}\left(t\right) & y_{S}\left(t\right) & z_{S}\left(t\right) \end{array}\right]}^{T}$ is the position vector of satellite S at any time t, $L_{V}\left(t\right)={\left[\begin{array}{ccc} x_{V}\left(t\right) & y_{V}\left(t\right) & z_{V}\left(t\right) \end{array}\right]}^{T}$ is the position coordinate vector of the vehicle under the frame of Earth-Centered-Earth-Fixed (ECEF)$,\;t_{V}^{S}\left(t\right)$ is time delay error between the receiver and satellite $and\;\varepsilon_{c}\left(t\right)$ is the correlated error, including the ephemeris error and the atmospheric error. It is assumed that the correlated errors are equal for different satellites if the localized vehicles are close. $\varepsilon_{u}\left(t\right)$ refer to the uncorrelated errors, e.g., thermal noise and multi-path error, which are hard to model because they are affected by the environment. The first-order Auto-Regression (AR) is the most popular model to describe the uncorrelated errors as shown in Equation (2).
(2)$$\varepsilon_{u}\left(t\right)=\{\begin{array}{ll} a\varepsilon_{u}\left(t-1\right)+n_{u}\left(t\right) & static\\ n_{u}\left(t\right) & dynamic \end{array}$$
where a is the dimensionless autoregression coefficient 0 or 1; $n_{u}\left(t\right)$ is a normally distributed random variable and obeys the Gaussian distribution, whose mean is zero and variance is an $\sigma_{u}^{2}$ i.e., $n_{u}\left(t\right)\sim \left(0,\sigma_{u}^{2}\right)$.

## 3. IVD Estimation Method

This section will present four IVD estimation methods under the cooperative localization framework, i.e., Absolute Position Differencing (APD), Pseudorangs Differencing (PD), Single Differencing (SD) and Double Differencing (DD). The parameters and nomenclature used in this section are listed in Table 1.

### 3.1. Absolute Position Differencing Distance

Connected vehicle technology enables communication and information-sharing between vehicles, allowing the vehicle to send its own position and receive the positions of its neighbors. Thus, we propose the first method to calculate the IVD through differencing the positions of vehicles directly. That is, the estimated IVD between ith and jth vehicle can be calculated
(3)$${\hat{D}}_{ij}\left(t\right)=||{\hat{L}}_{v_{i}}\left(t\right)-{\hat{L}}_{v_{j}}\left(t\right)||$$
where ${\hat{L}}_{v_{i}}\left(t\right)={\left[\begin{array}{ccc} x_{v_{i}}\left(t\right) & y_{v_{i}}\left(t\right) & z_{v_{i}}\left(t\right) \end{array}\right]}^{T}$ and ${\hat{L}}_{v_{j}}\left(t\right)={\left[\begin{array}{ccc} x_{v_{j}}\left(t\right) & y_{v_{j}}\left(t\right) & z_{v_{j}}\left(t\right) \end{array}\right]}^{T}$ are the estimated position vectors of vehicles $v_{i}$ and $v_{j}$, respectively. To improve the localization accuracy of each vehicle, the Weighted Least Square (WLS) is presented to optimize the vehicle’s position. The position and user clock offset are updated with multiple iterations until the solution converges according to a defined criterion. At iteration n, the estimated position ${\hat{L}}_{V}^{n}\left(t\right)$ is
(4)$${\hat{L}}_{V}^{n}\left(t\right)={\hat{L}}_{V}^{n-1}\left(t\right)+∆L_{V}^{n}$$
where ${\hat{L}}_{V}^{n-1}\left(t\right)$ is the position of the previous iteration and $∆L_{V}^{n}$ is the position increment at time t for current iteration n. $∆t_{V}^{n}$ refers to the clock deviation and $∆\varepsilon_{V}^{n}$ is the measured noise. Then, let $x={\left[\begin{array}{cc} ∆L_{V}^{n} & ∆t_{V}^{n} \end{array}\right]}^{T}$, $y={\left[\begin{array}{ccc} ∆\varepsilon_{V}^{1} & \cdots & ∆\varepsilon_{V}^{n} \end{array}\right]}^{T}$ be a set of noisy measurements that are linearly related to x, and the maximum likelihood estimate of x is defined as [34]
$$\hat{x}=arg\;\underset{x}{max}\;\frac{1}{{\left(2\pi \right)}^{\frac{N}{2}}{\left|R_{n}\right|}^{\frac{1}{2}}}\;e^{-\frac{1}{2}{\left(y-Hx\right)}^{T}R_{n}^{-1}\left(y-Hx\right)}$$
(5)$$=arg\;\underset{x}{min}{\left(y-Hx\right)}^{T}R_{n}^{-1}\left(y-Hx\right)\;$$
$$={\left(H^{T}R_{n}^{-1}H\right)}^{-1}H^{T}R_{n}^{-1}y$$
where H is the cosine matrix describing the measured value and $R_{n}$ is the covariance matrix associated with the measurement error. Figure 2 shows the process of the APD-based algorithm. As shown, by using the optimized absolute position of each vehicle, the IVD is calculated by following Equation (3).

### 3.2. Pseudoranges Differencing Distance

In addition to sharing the vehicles’ position, the vehicle can also share its pseudorange directly through V2V communication. Thus, this subsection uses the pseudoranges to calculate the IVD.

As shown in Figure 3, there are two vehicles $v_{i}$ and $v_{j}$, and one base station that can provide the accurate position information $L_{a}\left(t\right)={\left[x_{a}\left(t\right)\;y_{a}\left(t\right)\;z_{a}\left(t\right)\right]}^{T}$. The pseudoranges of the above three points are $\rho_{v_{i}}^{S}\left(t\right)$, $\rho_{v_{j}}^{S}\left(t\right)$ and $\rho_{a}^{S}\left(t\right)$. At any time t, the position of the vehicle $v_{j}$, $L_{v_{j}}\left(t\right)$, can be calculated by using the position of the vehicle $L_{v_{i}}\left(t\right)$, that is,
(6)$$L_{v_{j}}\left(t\right)=L_{v_{i}}\left(t\right)+D_{ij}\left(t\right)$$
where the vector $D_{ij}\left(t\right)={\left[\begin{array}{ccc} ∆x_{ij}\left(t\right) & ∆y_{ij}\left(t\right) & ∆z_{ij}\left(t\right) \end{array}\right]}^{T}$ is composed by the distances between vehicles in the x, y and z coordinates, respectively. The position of the vehicle $v_{i}$, the position of the vehicle $v_{j}$, and the satellite S have the following pseudorange relationship:(7)$$\rho_{v_{i}}^{S}\left(t\right)=||L_{S}\left(t\right)-L_{v_{i}}\left(t\right)||+t_{v_{i}}^{S}\left(t\right)+\varepsilon_{c}\left(t\right)+\varepsilon_{u_{i}}\left(t\right)$$
(8)$$\rho_{v_{j}}^{S}\left(t\right)=||L_{S}\left(t\right)-L_{v_{i}}\left(t\right)-D_{ij}\left(t\right)||+t_{v_{j}}^{S}\left(t\right)+\varepsilon_{c}\left(t\right)+\varepsilon_{u_{j}}\left(t\right)$$
(9)$$\rho_{a}^{S}\left(t\right)=||L_{S}\left(t\right)-L_{a}\left(t\right)||+t_{a}^{S}\left(t\right)+\varepsilon_{c}\left(t\right)+\varepsilon_{u_{a}}\left(t\right)$$

Then, the pseudorange differences between the vehicles and satellite are [33]
(10)$$\begin{array}{ll} ∆\rho_{ai} & =\rho_{a}^{S}\left(t\right)-\rho_{v_{i}}^{S}\left(t\right)=\sqrt{{\left(x_{s}\left(t\right)-x_{a}\left(t\right)\right)}^{2}+{\left(y_{s}\left(t\right)-y_{a}\left(t\right)\right)}^{2}+{\left(z_{s}\left(t\right)-z_{a}\left(t\right)\right)}^{2}}\\ & -\sqrt{{\left(x_{s}\left(t\right)-x_{v_{i}}\left(t\right)\right)}^{2}+{\left(y_{s}\left(t\right)-y_{v_{i}}\left(t\right)\right)}^{2}+{\left(z_{s}\left(t\right)-z_{v_{i}}\left(t\right)\right)}^{2}}+\left(t_{a}^{S}\left(t\right)-t_{v_{i}}^{S}\left(t\right)\right) \end{array}$$

Since
$$x_{v_{i}}\left(t\right)=x_{a}\left(t\right)+∆x_{ai}\left(t\right)\;y_{v_{i}}\left(t\right)=y_{a}\left(t\right)+∆y_{ai}\left(t\right)$$
(11)$$z_{v_{i}}\left(t\right)=z_{a}\left(t\right)-∆z_{ai}\left(t\right)$$
$$∆x_{ai}\left(t\right)=∆x_{aj}\left(t\right)-∆x_{ij}\left(t\right)\;∆y_{a_{i}}\left(t\right)=∆y_{aj}\left(t\right)-∆y_{ij}\left(t\right)\;∆z_{a_{i}}\left(t\right)=∆z_{aj}\left(t\right)-∆z_{ij}\left(t\right)$$

Equation (10) can be transformed into Equation (12) by using a Taylor series and the first-order partial derivatives to eliminate nonlinear terms [33].
(12)$$∆\rho_{ai}=\frac{x_{s}\left(t\right)-x_{a}\left(t\right)}{R_{a}^{S}\left(t\right)}∆x_{ai}\left(t\right)+\frac{y_{s}\left(t\right)-y_{a}\left(t\right)}{R_{a}^{S}\left(t\right)}∆y_{ai}\left(t\right)+\frac{z_{s}\left(t\right)-z_{a}\left(t\right)}{R_{a}^{S}\left(t\right)}∆z_{ai}\left(t\right)-∆t_{ai}\left(t\right)$$
with
$$R_{a}^{S}\left(t\right)=\sqrt{{\left(x_{s}\left(t\right)-x_{a}\left(t\right)\right)}^{2}+{\left(y_{s}\left(t\right)-y_{a}\left(t\right)\right)}^{2}+{\left(z_{s}\left(t\right)-z_{a}\left(t\right)\right)}^{2}}$$
(13)$$∆t_{ai}\left(t\right)=t_{v_{i}}\left(t\right)-t_{a}\left(t\right)$$
$$∆t_{aj}\left(t\right)=∆t_{a_{i}}\left(t\right)+∆t_{v_{i}}\left(t\right)+∆t_{v_{j}}\left(t\right)$$

It is assumed that both vehicles could observe the same number of satellites, i.e., S=1,2,3⋯N. Let
(14)$$a_{xS}=\frac{x_{s}\left(t\right)-x_{a}\left(t\right)}{R_{a}^{S}\left(t\right)}$$
(15)$$a_{yS}=\frac{y_{s}\left(t\right)-y_{a}\left(t\right)}{R_{a}^{S}\left(t\right)}$$
(16)$$a_{zS}=\frac{z_{s}\left(t\right)-z_{a}\left(t\right)}{R_{a}^{S}\left(t\right)}$$

Then,
(17)$$∆\rho =\psi ∆x=\left[\begin{array}{cc} \psi_{1} & 0_{N\times 4}\\ 0_{N\times 4} & \psi_{1} \end{array}\right]\left[\begin{array}{c} ∆x_{aj}\left(t\right)\\ ∆y_{aj}\left(t\right)\\ ∆z_{aj}\left(t\right)\\ ∆t_{ai}\left(t\right)\\ ∆x_{a_{j}}\left(t\right)\\ ∆y_{a_{j}}\left(t\right)\\ ∆z_{a_{j}}\left(t\right)\\ ∆t_{aj}\left(t\right) \end{array}\right]$$

Then, we can estimate the IVD between the two vehicles as
(18)$${\hat{D}}_{ij}\left(t\right)=∆{\hat{D}}_{aj}\left(t\right)-∆{\hat{D}}_{ai}\left(t\right)$$
where $∆{\hat{D}}_{aj}\left(t\right)$ and $∆{\hat{D}}_{ai}\left(t\right)$ are the estimated quantities of $∆D_{aj}\left(t\right)$ and $∆D_{ai}\left(t\right)$, respectively. That is,
(19)$$||{\hat{D}}_{ij}\left(t\right)||=\sqrt{∆{\hat{x}}_{ij}{\left(t\right)}^{2}+∆{\hat{y}}_{ij}{\left(t\right)}^{2}+∆{\hat{z}}_{ij}{\left(t\right)}^{2}}$$

In summary, Figure 4 shows the process of the PD-based algorithm. By using the pseudorange between the vehicle and the satellite, the pseudorange differences between localized vehicle and base station are calculated by following the geometry principle of Equation (17). Then, the IVD between two vehicles is computed based on the pseudorange differences.

### 3.3. Single Differencing Distance

The PD estimates the IVD by using the information of the base station; this subsection introduces the single differential method to calculate the IVD only using the pseudorange of vehicles. As shown in Figure 5, the SD method estimates the IVD by using the pseudorange of two vehicles from the same satellite. In this way, the clock difference between vehicle receivers can be eliminated, as well as the atmospheric delay error. In general, the atmospheric delay error includes ionospheric delay error and tropospheric delay error. When the GNSS satellite’s signal passes through the atmosphere, the signal will be affected by the electron density and water vapor density of the atmosphere. As a result, the signal propagation speed will change as well as the signal propagation time. Using the original signal propagation time to calculate the pseudorange will inevitably cause pseudorange errors.

As seen in Figure 5, since the satellite is far away from the vehicles, the pseudoranges of these two vehicles are assumed to be in parallel. Thus, the difference of the pseudoranges as [33]
(20)$$\begin{array}{c} l_{v_{i}v_{j}}^{S}\left(t\right)=\rho_{v_{i}}^{S}\left(t\right)-\rho_{v_{j}}^{S}\left(t\right)\\ =∆R_{V}^{S}\left(t\right)+∆t_{V}\left(t\right)+∆\varepsilon_{V}\left(t\right) \end{array}$$

As seen, the common errors are eliminated; however, the unusual error term $∆\varepsilon_{V}\left(t\right)$ increases. Since the ranges $R_{v_{i}}^{S}\left(t\right)$ and $R_{v_{j}}^{S}\left(t\right)$ are much larger than the distance between vehicles, we can estimate the difference of the pseudorange by using
(21)$$∆R_{V}^{S}\left(t\right)={\left[e^{S}\right]}^{T}\cdot D_{ij}\left(t\right)$$
where $e^{S}=\frac{L_{S}\left(t\right)-L_{v_{i}}\left(t\right)}{||L_{S}\left(t\right)-L_{v_{i}}\left(t\right)||}$ is the unit vector from vehicle to satellite. Assuming there are N satellites between the vehicles, we can have the following myopic values.
(22)$$\left[\begin{array}{c} l_{v_{i}v_{j}}^{1}\left(t\right)\\ l_{v_{i}v_{j}}^{2}\left(t\right)\\ \vdots \\ l_{v_{i}v_{j}}^{N}\left(t\right) \end{array}\right]\approx \left[\begin{array}{cc} {\left[e^{1}\right]}^{T} & 1\\ {\left[e^{2}\right]}^{T} & 1\\ \vdots & \vdots \\ {\left[e^{N}\right]}^{T} & 1 \end{array}\right]\left[\begin{array}{c} D_{ij}\left(t\right)\\ ∆t_{V}\left(t\right) \end{array}\right]$$

From the above equation, we can get the estimated value of $D_{ij}\left(t\right)$. Figure 6 shows the process of SD-based algorithm. First, the pseudorange difference between vehicles are obtained using Equation (20). Then, the IVD between vehicles $D_{ij}\left(t\right)$ is calculated by following Equation (22) with the triangle principle.

### 3.4. Double Differencing Distance

In addition to the SD, we can also use multiple satellites to achieve the IVD measurement. This subsection introduces the method combined the pseudorange information from two different satellites $S_{a}$ and $S_{b}$, as shown in Figure 7.

It is assumed that both vehicles can track the satellite $S_{a}$ and $S_{b}$, simultaneously. According to the SD method in Section 3.3, the difference in the pseudorange differences for different satellites is
(23)$$∆l_{v_{i}v_{j}}^{S_{a}S_{b}}\left(t\right)=l_{v_{i}v_{j}}^{S_{a}}\left(t\right)-l_{v_{i}v_{j}}^{S_{b}\;}\left(t\right)\;=∆r_{v_{i}v_{j}}^{S_{a}S_{b}}\left(t\right)+\varepsilon^{S_{a}S_{b}}\left(t\right)$$
where $∆r_{v_{i}v_{j}}^{S_{a}S_{b}}\left(t\right)=∆R_{V}^{S_{a}}\left(t\right)-∆R_{V}^{S_{b}}\left(t\right)$ and $\varepsilon^{S_{a}S_{b}}\left(t\right)=∆\varepsilon_{V}^{S_{a}}\left(t\right)-∆\varepsilon_{V}^{S_{b}}\left(t\right)$. The receiver clock is eliminated, but the error of the uncorrelated term still increases. Thus, we can have
(24)$$∆r_{v_{i}v_{j}}^{S_{a}S_{b}}\left(t\right)={\left[e^{S_{a}}-e^{S_{b}}\right]}^{T}D_{ij}\left(t\right)$$

According to Equation (24), the myopic distance and the relative position between vehicles can be calculated. For example, selecting a satellite as the reference satellite, the myopic solution of the double difference matrix can be deduced as follows.
(25)$$\left[\begin{array}{c} ∆l_{v_{i}v_{j}}^{S_{1}S_{0}}\left(t\right)\\ ∆l_{v_{i}v_{j}}^{S_{2}S_{0}}\left(t\right)\\ \vdots \\ ∆l_{v_{i}v_{j}}^{S_{N}S_{0}}\left(t\right) \end{array}\right]\approx \left[\begin{array}{cc} {\left[e^{1}-e^{0}\right]}^{T} & 1\\ {\left[e^{2}-e^{0}\right]}^{T} & 1\\ \vdots & \vdots \\ {\left[e^{N}-e^{0}\right]}^{T} & 1 \end{array}\right]D_{ij}\left(t\right)$$

Figure 8 shows the process of DD-based algorithm. Different from the SD method, the DD algorithm requires the pseudorange between vehicle and satellite. The myopic distance and the relative position between vehicles can be calculated according to Equation (24). Then, the distance between the vehicles can be calculated by Equation (25) with the myopic solution of the double difference matrix.

## 4. Experimental Results and Discussion

To evaluate the performance of the proposed four methods, this section will introduce several static or dynamic field tests.

### 4.1. Experiment Setup

This paper adopts two vehicle platforms for data collection, one unmanned ground vehicle (Figure 9a) and one passage car (Figure 9b). Both vehicles are equipped with NAV982 GNSS/INS receiver, which is configured to provide raw GNSS data at a rate of 5 Hz for vehicle localization. The receiver’s GNSS active antenna is installed on the roof of each vehicle. The raw data are stored in the Industrial Personal Computer. In addition, the two vehicles can share their localization information by using installed V2X communication devices. A base station (Figure 9c) is also installed in an open sky location, which can provide its accurate localization. The following sections will introduce the static and dynamic experimental studies and the analysis of their results.

### 4.2. Static Experiments

The static experiments were conducted in the parking lot of Southeast University, Nanjing. Six scenarios were tested, i.e., with the vehicle distances of 5 m, 10 m, 15 m, 25 m, 35 m and 50 m. Each scenario was tested for 15 min, and the IVD was measured online. In addition, the two vehicle platforms could observe the same GPS satellites. To evaluate the performance of different estimation methods quantitatively, the root means square error (*RMSE*) is proposed.
(26)$$RMSE=\sqrt{\frac{1}{T}\overset{T}{\underset{t=1}{\sum }}{\left[D\left(t\right)-\hat{D}\left(t\right)\right]}^{2}}$$
where D(t) is the true IVD between the two vehicles at time t, $\hat{D}\left(t\right)$ is the estimated IVD at time t, and T is the number of samples during the period.

Figure 10, Figure 11, Figure 12, Figure 13, Figure 14 and Figure 15 show the results of the static experiments. The weighted least-squares method is used to ensure the consistency of the vehicle spacing measurement for different algorithms. Table 2 gives the *RMSE* for different methods. The estimation accuracy is mainly affected by the correlated errors (ephemeris error, satellite clock error, atmospheric delay, etc.) and the uncorrelated errors (multipath effect and receiver thermal noise). In the static experiment, the uncorrelated errors are almost negligible because the tests are conducted in the open-sky condition.

As observed, the DD shows superior performance among all estimation methods in all tested scenarios. For example, the IVD errors of DD are smaller than 3 m in all six tested scenarios as shown in Figure 10, Figure 11, Figure 12, Figure 13, Figure 14 and Figure 15. The RMSE of DD are all smaller than 1 m, while those of other methods are larger than 1.5 m. This is mainly because the DD method could eliminate the correlated errors by using the differential approach, and the errors are minimized rapidly once the time synchronization is done. In addition, the performance of DD is stable with the increase in real IVD, e.g., 0.85 m for 5 m IVD and 0.75 m for 50 m IVD.

PD and SD methods can also reduce the correlated errors with the differential technique; however, their estimation errors may be amplified if uncorrelated errors exist. As observed from Table 2, the RMSE of PD is between 1.5–2 m, while RMSE of SD is between 2–3 m. Thus, PD has better accuracy than SD. Finally, the APD method shows the worst performance among all methods, i.e., with more than 3 m RMSE. Since it cannot eliminate the uncorrelated errors of pseudorange, the APD is the most unstable method.

### 4.3. Dynamic Experiments

The proposed IVD estimation methods should be applied to moving vehicles in real traffic conditions. Thus, this subsection will conduct several experiments for moving vehicles under different driving conditions.

The passenger car drove forward and pulled the unmanned ground vehicle to keep the constant IVD. The tested IVDs were 5, 10 and 15 m for the dynamic experiment. Two driving conditions were tested, i.e., the open-sky condition and the condition with roadside tree covering. Their trajectories are given in Figure 16a,b respectively.

Table 3, Figure 17 and Figure 18 show the measured IVD results for both two conditions. The estimated accuracy is different for different methods. In open-sky conditions, the DD method shows the best performance. The maximum error of DD was smaller than 3.5 m as observed in Figure 17, and its RMSE was around 1 m, which is similar to the results of the static experiment. Similarly, the errors of other methods follow the static experiment and are sorted by PD, SD and APD. This can be explained by the fact that the uncorrelated errors, mainly caused by multipath, are small. In addition, the maximum errors of APD sometimes are larger than the IVD, which may lead to collision if the estimated IVD is used in the vehicle control.

In the roadside tree covering condition (Figure 18), since the direct path of the GPS signal to the receiver is blocked by the trees, the uncorrelated errors caused by the multi-path increase. The multi-path may introduce time-varying deviation into the distance measurement; thus the IVD estimated by SD and DD methods may be amplified. As a result, the PD shows better performance than DD because the uncorrelated errors, i.e., multi-path and receiver thermal noise, cannot be eliminated by the differential technique. The RMSE of PD is around 2.5 m, while DD’s RMSE is three times the results of static experiments (around 3.5 m). It should be noted that time synchronization should be done to avoid the clock errors in the DD method. Meanwhile, the estimated errors of SD and APD methods also increase to 4 and 9 m due to the increase in the multi-path errors. APD is still the most unstable among all methods.

## 5. Conclusions

This paper develops four inter-vehicle distance estimation methods, i.e., APD, PD, SD and DD, based on cooperative vehicle localization. The vehicle’s absolute position or pseudorange are shared among vehicles by using the V2V communication devices. Static and dynamic experiments were conducted to evaluate and compare their performance. The results show that the DD method shows superior performance among the four methods if the uncorrelated errors are small or negligible (static experiment or dynamic experiment with open-sky condition). When the multi-path errors emerge due to the blocked GPS signal, the PD method using the original pseudorange is more effective because the uncorrelated errors cannot be eliminated by the differential technique. In addition, the accuracy of the DD method may be worse if the uncorrelated errors increase. In dynamic experiments, two different types of field scenarios are reported, i.e., open-sky and roadside tree covering. The results show that the DD method is only superior in an open sky environment since it may be more sensitive to the multipath effect. Using raw pseudorange for IVD estimation is more effective if the multipath effect is observed.

Several future works are planned. First, the effect of communication range, capacity and delay on cooperative localization will be investigated. Second, the speed of moving vehicles will affect the performance of the proposed cooperative localization framework. A robust method to combine more information may be required to increase the IVD estimation accuracy. Finally, a novel approach that can address the loss of GNSS signal under extreme environments such as urban forest will also be studied in the future.

## Figures and Tables

**Figure 1 sensors-21-02048-f001:**
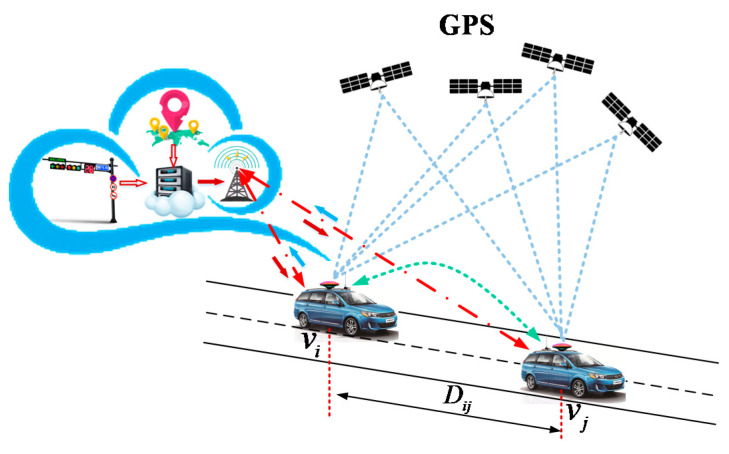
Cooperative localization by using multi-source information fusion.

**Figure 2 sensors-21-02048-f002:**
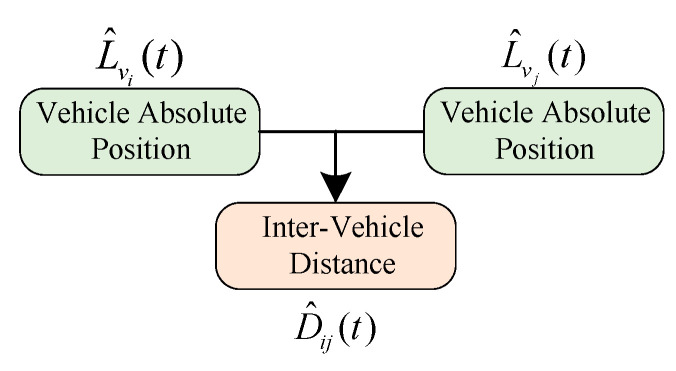
Flowchart of Absolute Position Differencing (APD)-based inter-vehicle distance (IVD) estimation method.

**Figure 3 sensors-21-02048-f003:**
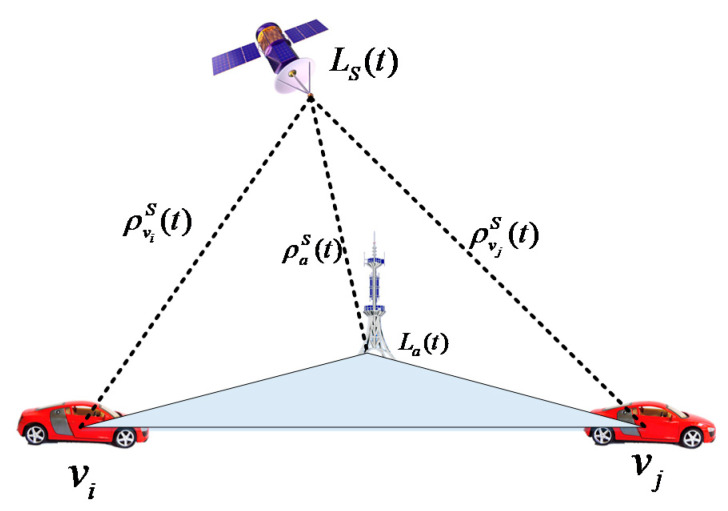
Pseudoranges differencing distance.

**Figure 4 sensors-21-02048-f004:**
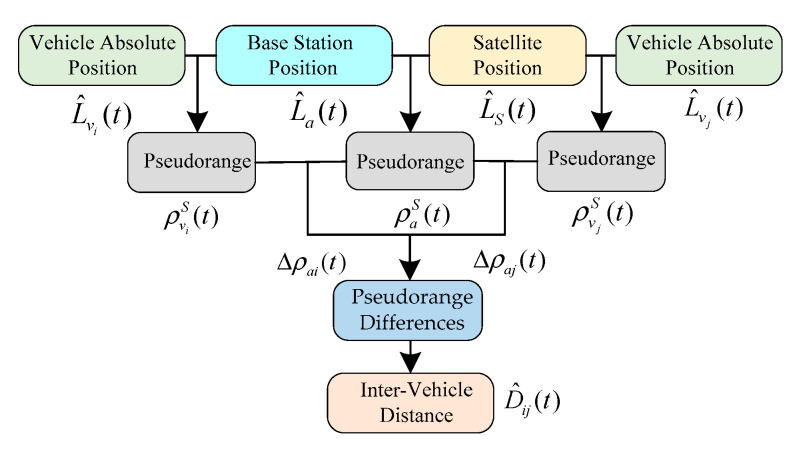
Flowchart of Pseudorange Differencing (PD)-based IVD estimation method.

**Figure 5 sensors-21-02048-f005:**
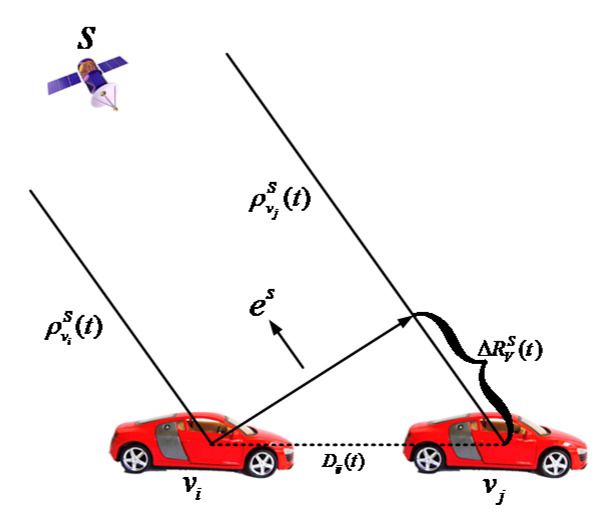
Single differencing distance.

**Figure 6 sensors-21-02048-f006:**
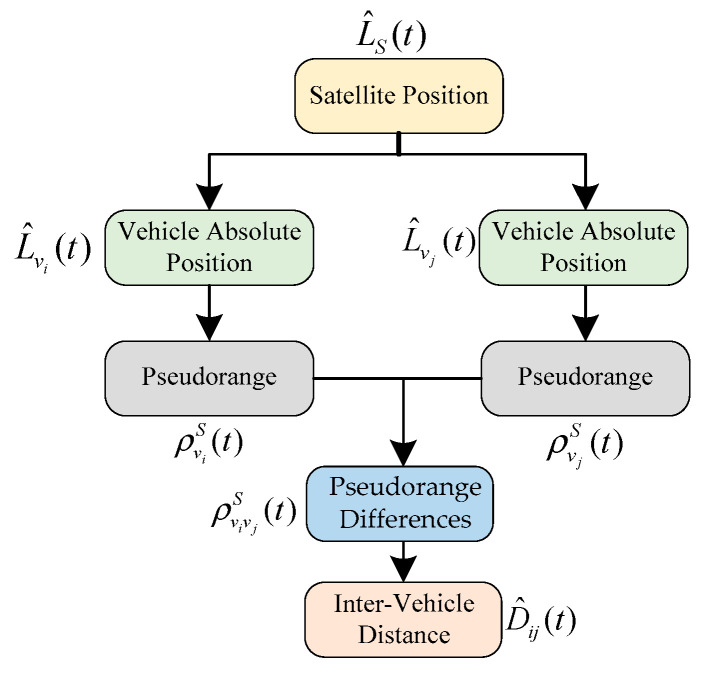
Flowchart of Single Differencing (SD)-based IVD estimation method.

**Figure 7 sensors-21-02048-f007:**
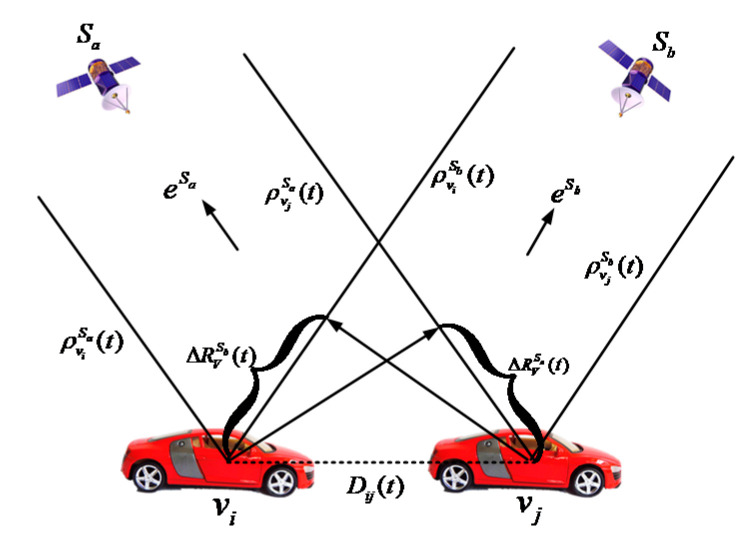
Double differencing distance.

**Figure 8 sensors-21-02048-f008:**
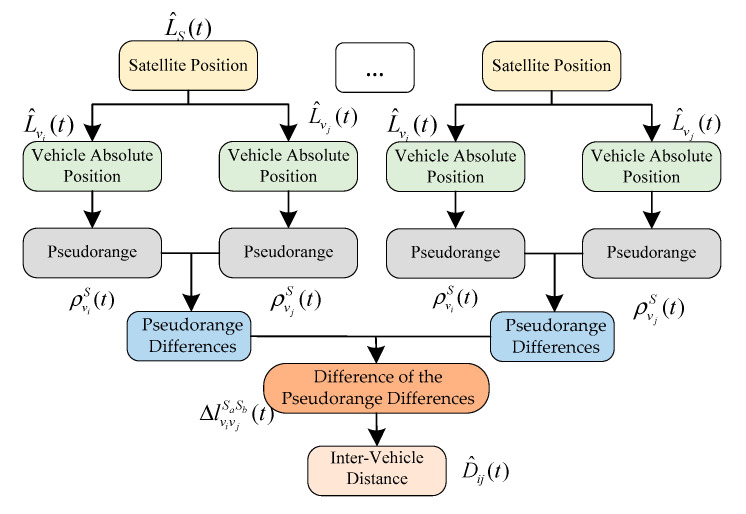
Flowchart of double difference (DD)-based IVD estimation method.

**Figure 9 sensors-21-02048-f009:**
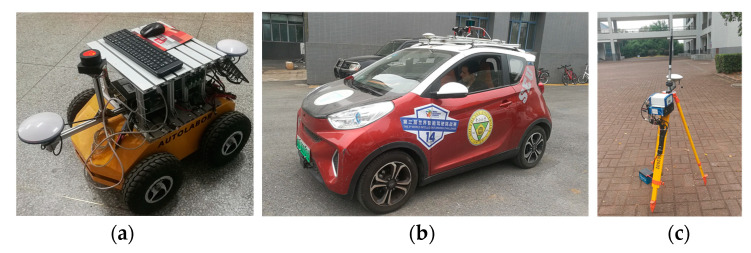
Experiment equipment: (**a**) Unmanned ground vehicle; (**b**) Passenger car; (**c**) Base station.

**Figure 10 sensors-21-02048-f010:**
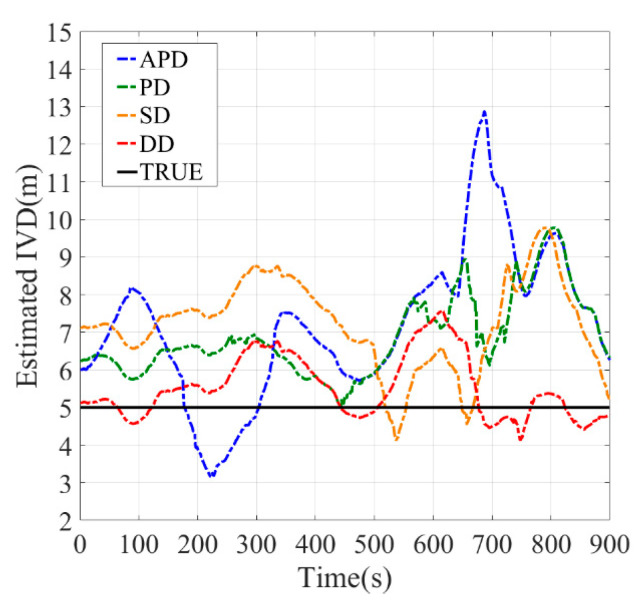
Baseline = 5 m.

**Figure 11 sensors-21-02048-f011:**
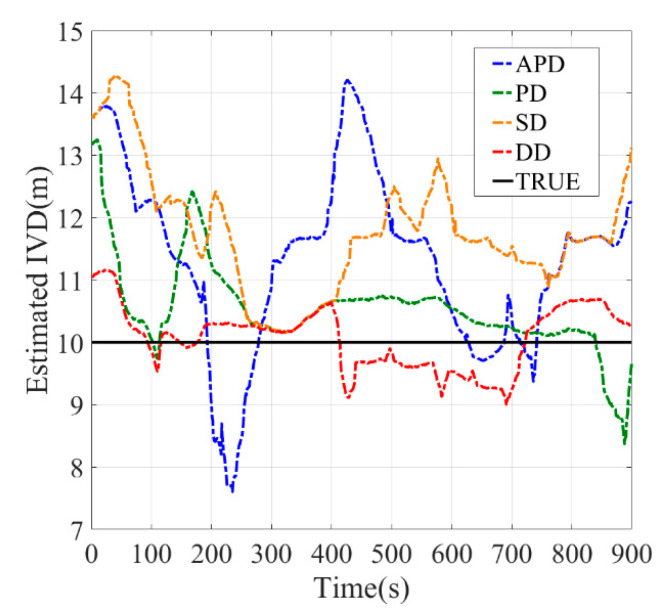
Baseline = 10 m.

**Figure 12 sensors-21-02048-f012:**
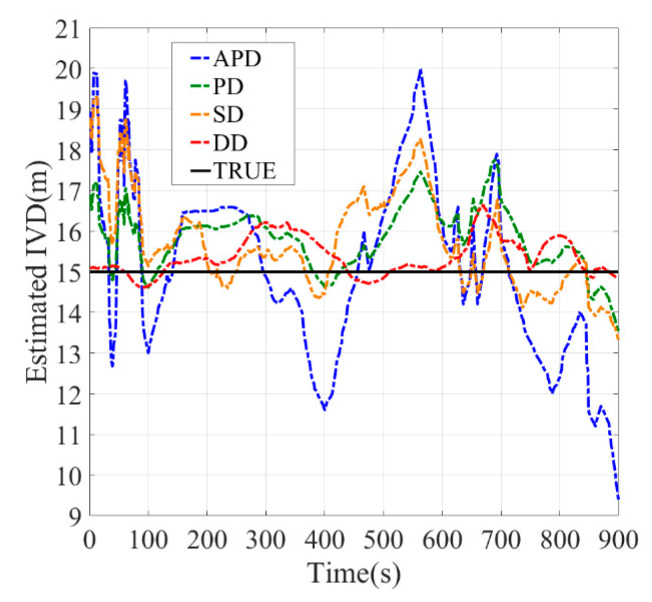
Baseline = 15 m.

**Figure 13 sensors-21-02048-f013:**
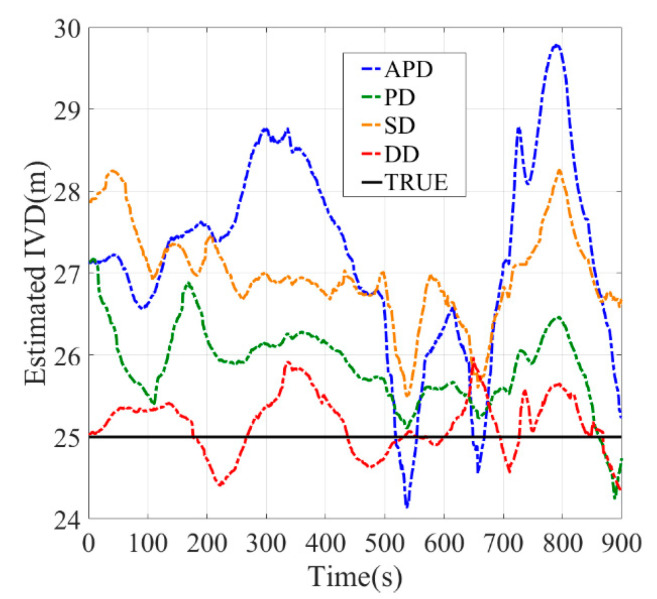
Baseline = 25 m.

**Figure 14 sensors-21-02048-f014:**
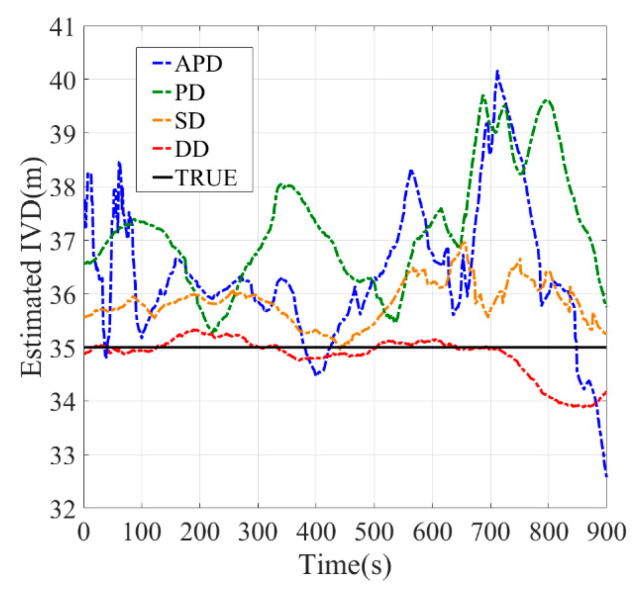
Baseline = 35 m.

**Figure 15 sensors-21-02048-f015:**
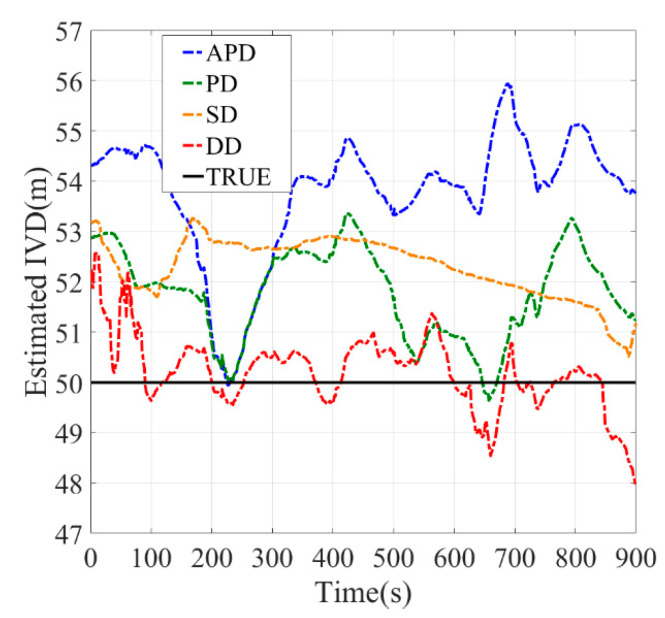
Baseline = 50 m.

**Figure 16 sensors-21-02048-f016:**
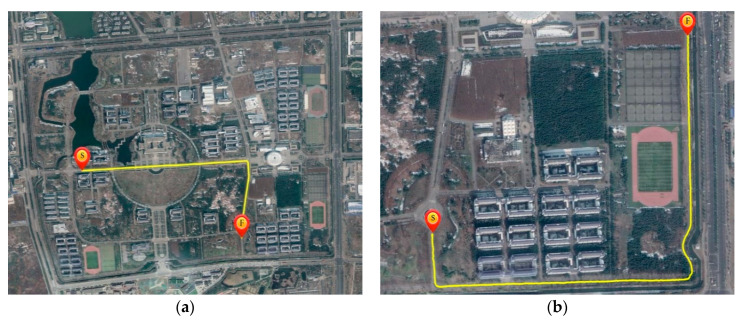
Experiment environment, (**a**) Open-sky condition, (**b**) Condition with roadside tree covering.

**Figure 17 sensors-21-02048-f017:**
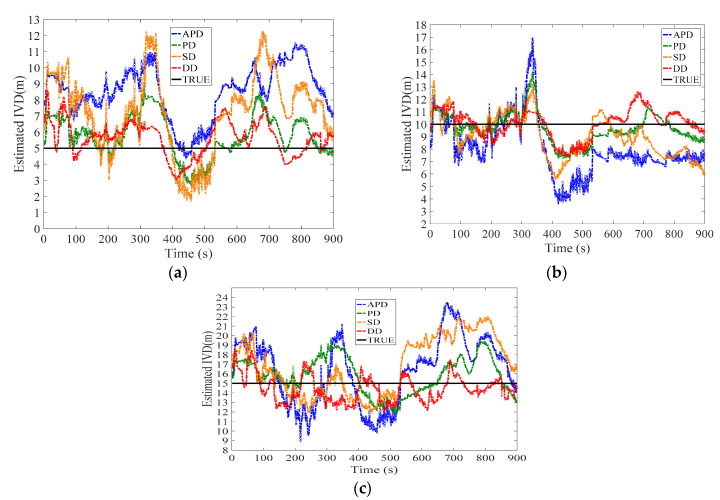
Dynamic experiments with open sky scenario, (**a**) Baseline = 5 m, (**b**) Baseline = 10 m, (**c**) Baseline = 15 m.

**Figure 18 sensors-21-02048-f018:**
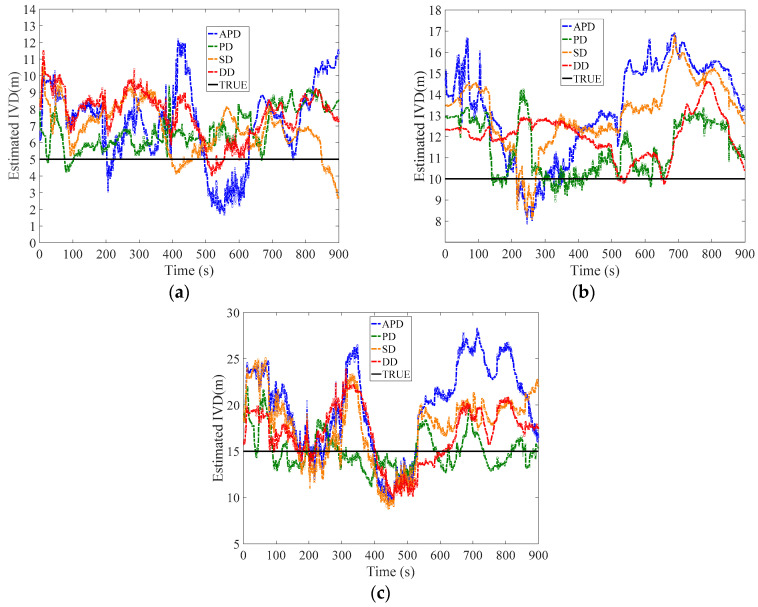
Dynamic experiments with roadside tree covering condition, (**a**) Baseline = 5 m, (**b**) Baseline = 10 m, (**c**) Baseline = 15 m.

**Table 1 sensors-21-02048-t001:** Nomenclature used in this paper.

Notation	Description	Notation	Description
$D_{ij}\left(t\right)$	Vehicle distance vector	H	Cosine matrix
${\hat{D}}_{ij}\left(t\right)$	Estimated IVD between ith and jth vehicle	$\varepsilon_{c}\left(t\right)$	Correlated errors
$e^{S}$	Unit vector from vehicle to satellite	$\varepsilon_{u}\left(t\right)$	Uncorrelated errors
$L_{a}\left(t\right)$	Position of base station	$∆\varepsilon_{V}\left(t\right)$	Unusual error term
$L_{V}\left(t\right)$	Position of Vehicle V	x	Maximum likelihood estimate
${\hat{L}}_{V}^{n-1}\left(t\right)$	Estimated position of vehicle V for previous iteration	$t_{V}^{S}\left(t\right)$	Time delay error between the receiver and satellite
${\hat{L}}_{V}\left(t\right)$	Estimated position of vehicles V	$∆t_{V}\left(t\right)$	Difference of time delay error
$l_{v_{i}v_{j}}^{S}\left(t\right)$	Pseudorange difference between ith and jth vehicle	$\rho_{V}^{S}\left(t\right)$	Pseudorange from vehicle V to satelli S
$∆L_{V}^{n}$	Position increment	$∆\rho_{ai}$	Pseudorange differences for the vehicle i and base station a
$∆l_{v_{i}v_{j}}^{S_{a}S_{b}}\left(t\right)$	Pseudorange difference between ith and jth vehicle for different satellites	$R_{V}^{S}\left(t\right)$	True distance between vehicle V and satellite S

**Table 2 sensors-21-02048-t002:** The RMSE for different methods in the static experiment.

Ture IVD	APD RMSE [m]	PD RMSE [m]	SD RMSE [m]	DD RMSE [m]
5 m	3.93	1.98	2.52	0.85
10 m	3.65	1.75	2.43	0.91
15 m	2.53	1.68	2.39	0.98
25 m	3.24	1.53	2.74	1.03
35 m	3.43	1.64	1.02	0.42
50 m	3.95	1.99	2.39	0.75

**Table 3 sensors-21-02048-t003:** The RMSE of all methods in dynamic experiment.

(**a**) Open-Sky Condition.				
**Ture IVD**	**APD RMSE [m]**	**PD RMSE [m]**	**SD RMSE [m]**	**DD RMSE [m]**
5 m	3.14	0.98	2.52	0.85
10 m	3.27	1.25	2.05	1.14
15 m	3.53	1.68	2.39	1.25
(**b**) Roadside Tree Covering Condition.				
**Ture IVD**	**APD RMSE [m]**	**PD RMSE [m]**	**SD RMSE [m]**	**DD RMSE [m]**
5 m	3.86	1.98	2.82	2.35
10 m	5.65	2.29	4.01	3.05
15 m	7.64	3.28	5.97	3.84

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
