# Peer review of "Robust Inter-Vehicle Distance Measurement Using Cooperative Vehicle Localization"

_sensors, 2021, doi:10.3390/s21062048_

Round 1

Reviewer 1 Report

The contribution of this manuscript has not been clearly presented in the abstract.

The introduction presented too way broad topics about localization algorithms, even mentioning SLAM. The authors should focus on more relevant existing studies on estimating the inter-vehicle distance (IVD). In the introduction as well, the original contribution of the proposed method comparing with existing algorithms has not been clearly presented.

The English presentation should be polished throughout the whole manuscript. The current style can be prevent understanding the proposed method smoothly.

There are no descriptions of Algorithm 1, 2, 3, and 4.

The authors said that the weighted least squares method was employed to reduce the pseudo-range errors and noises. The authors should clearly explain how the weighted least squares method works on the four IVD algorithms. It seems that the weighted least squares method just was used as an evaluation criterion. It is hard to value the use of a well-known evaluation criterion as technical contributions.

Author Response

Dear Editors and Reviewer,

Thank you for the letter and reviewer’ comments concerning out manuscript entitled “Robust Inter-Vehicle Distance Measurement Using Cooperative Vehicle Localization” (ID: Sensors-1106994). Those comments are all valuable and very helpful for revising and improving our paper, as well as the critical guiding significance to our researches. We have carefully reviewed, adopted most reviewer’ suggestions and revised this paper. The revised portion is marked in blue in the paper. The main revision and response to the reviews’ comments are as following:

Responds to the reviewer’ comments:

Reviewer #1:

1. The contribution of this manuscript has not been clearly presented in the abstract.

Response: Thank you for this suggestion. We have added the main contributions of this paper in the Abstract and marked in blue. Thus, this paper proposes a cooperative localization framework which shares the absolute position or pseudorange by using V2X communication devices to estimate the IVD between vehicles. Four IVD estimation methods are presented, Absolute Position Differencing (APD), Pseudorange Differencing (PD), Single Differencing (SD) and Double Differencing (DD). Several static and dynamic experiments are conducted to evaluate and compare their measurement accuracy.

2. The introduction presented too way broad topics about localization algorithms, even mentioning SLAM. The authors should focus on more relevant existing studies on estimating the inter-vehicle distance (IVD). In the introduction as well, the original contribution of the proposed method comparing with existing algorithms has not been clearly presented.

Response: Thank you for this suggestion. We are agreeing with the reviewer that this paper should focus on the introduction of IVD studies. We mentioned the SLAM technique in the background (only once time) and shifted to IVD discussion. We don’t think this will affect the core idea of this paper, which is the investigation of IVD. To avoid any misunderstanding, we added more literatures while discussing IVD.

Tomic empolyed the maximum likelihood convex optimization method [R1], and Naseri used Bayesian estimation method to improve the accuracy of the distance between two points [R2]. In addition, Guo achieved an infrastructure-free cooperative relative localization by using onboard ultra-wideband ranging and communication network [R3].

[R1] Tomic, S.; Beko, M.; Tuba, M., A Linear Estimator for Network Localization Using Integrated RSS and AOA Measurements. IEEE Signal Processing Letters 2019, 26, (3), 405-409.

[R2] Naseri, H.; Koivunen, V., A Bayesian Algorithm for Distributed Network Localization Using Distance and Direction Data. IEEE Transactions on Signal and Information Processing over Networks 2019, 5, (2), 290-304.

[R3] Guo, K.; Li, X.; Xie, L., Ultra-wideband and Odometry-Based Cooperative Relative Localization With Application to Multi-UAV Formation Control. IEEE Trans Cybern 2019.

In addition, for the concern ‘the original contribution of the proposed method comparing with existing algorithms has not been clearly’, we have revised the contribution of this paper. Comparing existing algorithms, we proposed several cheap methods, which only use the pseudorange to enhance the IVD estimation without any other device requirement. In addition, several experiments are conducted to verify the effectiveness of proposed methods.

The main contributions of this paper are twofold. First, four different IVD estimation framework are formulated and compared. The weighted least squares method is employed to reduce the pseudorange errors and noises. The correlation errors of pseudoranges (i.e., satellite clock error, satellite ephemeris error, ionospheric error and tropospheric error) are greatly reduced. Second, field experiments, including static and dynamic, open-sky and GNSS-blocked driving scenarios are conducted to verify their effectiveness. Among these methods, DD indicates the highest IVD measurement accuracy in open sky conditions, while PD shows the best accuracy in urban driving conditions with shelter.

3. The English presentation should be polished throughout the whole manuscript. The current style can be prevent understanding the proposed method smoothly.

Response: Thanks for this comment and suggestion. We have carefully checked the full text and corrected some simple errors, such as pictures, typos, grammar and references.

In addition, we were planning to send the paper to Language Editing Agent for English revision. Unfortunately, it may take more than 7 business days to complete the revision that the revision submission will be due.

We will ask a native speaker to revise the paper next time once all the technical issues are addressed. Thank you for your understanding.

4. There are no descriptions of Algorithm 1, 2, 3, and 4.

Response: Thank you for this suggestion. For any misunderstanding, we added the descriptions of each algorithm in the paper.

Figure 2 shows the process of APD-based algorithm. As shown, by using the optimized absolute position of each vehicle, the IVD is calculated by following Eq. (3).

In summary, Figure 4 shows the process of PD-based algorithm. By using the the pseudorange between the vehicle and the satellite, the pseudorange differences between localized vehicle and base station are calculated by following geometry principle as Eq. (17). Then, the IVD between two vehicles are computed based on the pseudorange differences.

Figure 6 shows the process of SD-based algorithm. First, the pseudorange difference between vehicles are obtained by using Eq. (20). Then, the IVD between vehicles D_ij (t) is calculated by following Eq. (22) with the triangle principle.

Figure 8 shows the process of DD-based algorithm. Different from SD method, DD algorithm requires the pseudorange between vehicle and satellite. The myopic distance and the relative position between vehicles can be calculated according to Eq. (24). Then, the distance between the vehicles can be calculated by following Eq. (25) with the myopic solution of double difference matrix. The authors said that the weighted least squares method was employed to reduce the pseudo-range errors and noises.

5. The authors should clearly explain how the weighted least squares method works on the four IVD algorithms. It seems that the weighted least squares method just was used as an evaluation criterion. It is hard to value the use of a well-known evaluation criterion as technical contributions. Response: Thank you for this comment. The weighted least squares was used to reduce the pseudo-range errors and noises of each vehicle. That is, the pseudorange of each vehicle is optimized with the weighted least squares. And the four proposed methods will use the optimized pseudorange for IVD estimation. Thus, the weighted least squares method will improve the estimation accuracy of the four IVD algorithms. To avoid any misunderstanding, we have revised the statement,’ The weighted least squares method is employed to reduce the pseudorange errors and noises of each vehicle.’

All changes are highlighted and marked in blue. Thanks again for this comment. Finally, the authors are appreciating for the reviews’ and editors’ comments and suggestions.

Best Regards

Guodong Yin

Reviewer 2 Report

Conceptually, this paper is still comparable to the paper: "Muhammad Tahir; Saved Saad Afzal, Muhammad Saad Chughtai and Khurram Ali: On the Accuracy of Inter-Vehicular Range Measurements Using GNSS Observables in a Cooperative: IEEE Transactions on Intelligent Transportation Systems (Volume: 20, Issue: 2, Feb. 2019)" https://ieeexplore.ieee.org/author/38467085100"

In relation to the work reviewed (https://susy.mdpi.com/user/review/review/15506214/su8OhAFD) , the authors made corrections but not completely. They made a graphical representation but still kept a large number of equations that were downloaded but with different notations.

In line 340 should be next to the literature on the 16th put a capital letter.

Please do double check the literature, methodological errors and text editing.

Thank you to the authors for making the appropriate corrections.

Reviewer 3 Report

The authors deal with precise localization relative to safety for connected and automated vehicle (CAV). The paper proposes a cooperative localization framework which shares the absolute position or pseudorange by using V2X communication devices to estimate the inter-vehicle distance (IVD) between vehicles.

  • the title is fine, abstract and key words are appropriated,
  • suggestion, split chapter 1. Introduction to 1. Introduction and 2 Lliterature review,
  • Figure 3. better marking vehicle letters vi and vj, in picture,
  • row 176, explanation of atmospheric delay error,
  • research methods are appropriated,
  • chapter 5. Conclusion, must be spreaded.

Author Response

Dear Editors and Reviewer,

Thank you for the letter and reviewer’ comments concerning out manuscript entitled “Robust Inter-Vehicle Distance Measurement Using Cooperative Vehicle Localization” (ID: Sensors-1106994). Those comments are all valuable and very helpful for revising and improving our paper, as well as the critical guiding significance to our researches. We have carefully reviewed, adopted most reviewer’ suggestions and revised this paper. The revised portion is marked in blue in the paper. The main revision and response to the reviews’ comments are as following:

Responds to the reviewer’ comments:

  • Reviewer #3

The authors deal with precise localization relative to safety for connected and automated vehicle (CAV). The paper proposes a cooperative localization framework which shares the absolute position or pseudorange by using V2X communication devices to estimate the inter-vehicle distance (IVD) between vehicles.

  1. The title is fine, abstract and key words are appropriated, suggestion, split chapter 1. Introduction to 1. Introduction and 2 Literature review.
  • Response: Thank you for this comment. We are agreeing with the reviewer that splitting Section 1 to two sections may make this paper look better. However, to keep the smooth description of this paper, we think holding the background and motivations of this paper in one section is better. In addition, we have added some more literatures to introduce the studies related with IVD estimation. Listed literature are as follows:
  • [R1] Tomic, S.; Beko, M.; Tuba, M., A Linear Estimator for Network Localization Using Integrated RSS and AOA Measurements. IEEE Signal Processing Letters 2019, 26, (3), 405-409.
  • [R2] Naseri, H.; Koivunen, V., A Bayesian Algorithm for Distributed Network Localization Using Distance and Direction Data. IEEE Transactions on Signal and Information Processing over Networks 2019, 5, (2), 290-304.
  • [R3] Guo, K.; Li, X.; Xie, L., Ultra-wideband and Odometry-Based Cooperative Relative Localization With Application to Multi-UAV Formation Control. IEEE Trans Cybern 2019.
  1. Figure 3. better marking vehicle letters viand vj, in picture,
  • Response: Thank you for this comment. We have made changes, as shown in the figure below, and also made relevant changes in the manuscript.

Figure 2. Pseudoranges differencing distance

  1. Row 176, explanation of atmospheric delay error.
  • Response: Thank you for this comment. In general, the atmospheric delay error includes ionospheric delay error and tropospheric delay error. When the GNSS satellite’s signal passes through the atmosphere, the signal will be affected by the electron density and water vapor density of the atmosphere. As a result, the signal propagation speed will change as well as the signal propagation time. Using the original signal propagation time to calculate the pseudorange will inevitably cause pseudorange errors.
  • We have added the corresponding description in the paper.

  1. Research methods are appropriated, chapter 5. Conclusion, must be spreaded.
  • Response: Thank you for this comment. We have added more in the conclusion section.
  • In dynamic experiments, two different types of field scenarios have been reported: one open-sky and one roadside tree covering experiments. The results obtained in all experiments strongly support the observation that DD method is only superior in open sky environment. However, if the GNSS satellite’s signal is affected by multipath phenomenon strongly, using raw pseudorange is more effective compared to DD.

Finally, the authors are appreciating for the reviews’ and editors’ comments and suggestions.

Best Regards

Guodong Yin

Round 2

Reviewer 1 Report

The authors have appropriately discussed most of the issues raised in the last review.

Author Response

Dear Reviewer, Thank you for the letter and reviewers’ comments concerning out manuscript entitled “Robust Inter-Vehicle Distance Measurement Using Cooperative Vehicle Localization” (ID: Sensors-1106994). Those comments are all valuable and very helpful for revising and improving our paper, as well as the critical guiding significance to our researches. We have carefully reviewed, adopted most reviewer’ suggestions and revised this paper. The revised portion is marked in dark green in the paper. The main revision and response to the reviews’ comments are as following: Responds to the editors’ comments: The simplification that the authors introduce is quite common in literature. Even this is not the best solution, it can work in normal driving (results must be studied carefully and it could not be valid for critical situations). Please, introduce a mention to limitations of the model used.  Response: Thanks for your valuable suggestion. We have added related description in Section 2 Problem formulation and Section 5 Conclusion, which are highlighted in blue.  1) In the Section 2: Note that, since the signal of GNSS maybe lost in the environments of building blockings and tunnels, the critical situations which can severely reduce the complement of GNSS signal are not considered.  2) In the future work of Conclusion: Besides, the novel approach which can address the loss of GNSS signal under the extreme environment such as urban forest will also research in the future.  We have rechecked the paper to make sure all mistakes modified. Thanks again for this comment. Best regards, Guodong Yin

This manuscript is a resubmission of an earlier submission. The following is a list of the peer review reports and author responses from that submission.

Round 1

Reviewer 1 Report

It is clear that precise localization is critical to safety for connected and automated vehicle (CAV) and some recent developed CAV applications require accurate measurement of the inter-vehicle distance (IVD). This paper proposes a cooperative localization framework which shares the absolute position or pseudorange by using V2X communication devices to estimate the IVD between vehicles. The paper has been well presented. However, I think this paper needs Minor Revision before publication.

Comments:

(1) P2, line 50. In addition to single vehicle localization, some safety critical CAV applications, e.g., vehicle platooning, require accurate measurement of the inter-vehicle distance (IVD).

"Platooning" may be a misspelled word. It is recommended to check the spelling of the full text carefully.

(2) In order to express the calculation principle of the algorithm more clearly, it is recommended to give an algorithm flow chart before introducing the proposed algorithm in detail.

(3) The true value of the inter-vehicle distance for the dynamic experiment is 10 meters. It is recommended to add more different vehicle distances to verify the dynamic characteristics and reliability of the system.

(4) How the result of the experiment to support the "robust" of the title. For example, does dynamic experiments need to explore the impact of vehicle speed on the proposed algorithm?

Author Response

Dear Editors and Reviewers,

Thank you for the letter and reviewers’ comments concerning out manuscript entitled “Robust Inter-Vehicle Distance Measurement Using Cooperative Vehicle Localization” (ID: Sensors-1055599). Those comments are all valuable and very helpful for revising and improving our paper, as well as the critical guiding significance to our researches. We have carefully reviewed, adopted most reviewer’ suggestions and revised this paper.

Finally, the authors are appreciating for the reviews’ and editors’ comments and suggestions.

Best Regards

Guodong Yin

Reviewer 2 Report

The authors explore in the area of measuring distance between vehicles and as indicated that is an area of significant interest.
In their work, they referred to 36 titles (list of reference), but some literature is missing, such as:
Muhammad Tahir; Saved Saad Afzal, Muhammad Saad Chughtai and Khurram Ali: On the Accuracy of Inter-Vehicular Range Measurements Using GNSS Observables in a Cooperative: IEEE Transactions on Intelligent Transportation Systems (Volume: 20, Issue: 2, Feb. 2019) https://ieeexplore.ieee.org/author/38467085100
The list of reference must be supplemented (row 71-81).
Also in row71 the sentence begin with the lowercase letter. It needs to be corrected
“….. errors were small and no multipath error existed. their work is only simulation based which has not 71 been tested in real field experiments“
Also in FIg 1. authors use lowercase letter.
Authors must double-check the entire text.
The explanation of certain parameters in the equations, since there are 33 of them, are not noticeable or clearly defined. Also a good portion of the equations are taken from the literature so it may not be necessary for the authors to use all the equations. Perhaps it is enough to just refer to the literature and indicate the basic equations.
Figures 6-11 need to be briefly explained. The authors indicated that these are six scenarios where the distances are 5 m, 10 m, 15 m, 20, m, 35 m and 50 m. It is necessary to analyze what they observed at individual distances, compare the data and comment on them briefly. A similar need to be done for Figures 13-14.
In conclusion, it is necessary to give recommendations for further work. It is necessary that the results of the research contribute to energy efficiency (they mentioned it at the beginning of the paper).
Basically, this work is almost the same as that paper (conceptually) :
Muhammad Tahir; Saved Saad Afzal, Muhammad Saad Chughtai and Khurram Ali: On the Accuracy of Inter-Vehicular Range Measurements Using GNSS Observables in a Cooperative: IEEE Transactions on Intelligent Transportation Systems (Volume: 20, Issue: 2, Feb. 2019) https://ieeexplore.ieee.org/author/38467085100).

Where is the contribution of the authors in paper (what is new...)“Robust Inter-Vehicle Distance Measurement Using Cooperative Vehicle Localization“ ?
The paper should be completely reorganized with an indication of what has already been explored in the existing literature and what is the author's contribution to the paper.

Author Response

(The authors gave the same response as above.)

Round 2

Reviewer 2 Report

I thank the authors for their revised paper.

The authors have made significant improvements and have given a clear indication of the results of research activities.

Conceptually, this paper is still comparable to the paper:"Muhammad Tahir; Saved Saad Afzal, Muhammad Saad Chughtai and Khurram Ali: On the Accuracy of Inter-Vehicular Range Measurements Using GNSS Observables in a Cooperative: IEEE Transactions on Intelligent Transportation Systems (Volume: 20, Issue: 2, Feb. 2019) https://ieeexplore.ieee.org/author/38467085100"

 to which the author (s) now refer.

In their paper authors  still use a large number of equations that are downloaded, just with other tags. They now have a different meaning because they follow algorithms 1-4. Authors should supplement algorithms 1-4 with a graphical representation (loop).

It is necessary to double-check the literature that has been replaced (they left the old literature labels from the first paper).In lines 69 - 93 they indicated the modified literature, but in the rest of the paper it does not follow the indicated.

Author Response

Dear Reviewer,

Thank you for the letter and reviewers’ comments concerning out manuscript entitled “Robust Inter-Vehicle Distance Measurement Using Cooperative Vehicle Localization” (ID: Sensors-1055599). Those comments are all valuable and very helpful for revising and improving our paper, as well as the critical guiding significance to our researches. We have carefully reviewed, adopted most reviewer’ suggestions and revised this paper.

Finally, the authors are appreciating for the reviews’ and editors’ comments and suggestions.

Best Regards

Guodong Yin
